# Isolation of Unstable Isomers of Lucilactaene and Evaluation of Anti-Inflammatory Activity of Secondary Metabolites Produced by the Endophytic Fungus *Fusarium* sp. QF001 from the Roots of *Scutellaria baicalensis*

**DOI:** 10.3390/molecules25040923

**Published:** 2020-02-19

**Authors:** Sailesh Maharjan, Sang Bong Lee, Geum Jin Kim, Sung Jin Cho, Joo-Won Nam, Jungwook Chin, Hyukjae Choi

**Affiliations:** 1College of Pharmacy, Yeungnam University, Gyeongsan 38541, Korea; msaileshmaharjan@gmail.com (S.M.); kimgeumjin@naver.com (G.J.K.); jwnam@yu.ac.kr (J.-W.N.); 2Korea Institute of Medical Microrobotics, Gwangju 61011, Korea; sangbongyi1@kimiro.re.kr; 3New Drug Development Center, Daegu-Gyeongbuk Medical Innovation Foundation, Daegu 41061, Korea; sjcho@dgmif.re.kr

**Keywords:** endophytes, *Fusarium*, *Scutellaria baicalensis*, anti-inflammatory, lucilactaene

## Abstract

The filamentous fungal pathogen *Fusarium* sp. causes several crop diseases. Some *Fusarium* sp. are endophytes that produce diverse valuable bioactive secondary metabolites. Here, extensive chemical investigation of the endophytic fungus, *Fusarium* sp. QF001, isolated from the inner rotten part of old roots of *Scutellariae baicalensis* resulted in the isolation of two new photosensitive geometrical isomers of lucilactaene (compounds **2** and **3**) along with lucilactaene (**6**) and six other known compounds (fusarubin (**1**), (+)-solaniol (**4**), javanicin (**5**), 9-desmethylherbarine (**7**), NG391 (**8**) and NG393 (**9**)). Newly isolated isomers and lucilactaene were unstable under light at room temperature and tended to be a mixture in equilibrium state when exposed to a polar protic solvent during reversed phase chromatography. Normal phase chromatography under dim light conditions with an aprotic mobile phase led to the successful isolation of the relatively unstable isomers **2** and **3**. Their structures were elucidated as 8(Z)-lucilactaene (**2**) and 4(*Z*)-lucilactaene (**3**) based on spectroscopic data. The absolute configuration of **4** was speculated to be *R* by computer-assisted specific rotation analysis. The isolated compounds could inhibit NO production and suppress pro-inflammatory cytokines expression in LPS-stimulated macrophage cells. These properties of the isolated compounds indicate their potential use as anti-inflammatory drugs.

## 1. Introduction

Plants are a major source of bioactive compounds that play crucial roles in treatment of various diseases. However, in recent years, microorganisms associated with plants have proven to be a reservoir of novel compounds with pharmacological activities [1]. These microorganisms, commonly known as endophytes, spend the whole or part of its life cycle colonizing inter- and/or intra-cellularly inside the healthy tissues of the host plant, typically causing no apparent symptoms of disease [2,3]. However, these endophytes have received considerable attention in the last few years owing to their function of protecting plants against harsh environment, pest pathogens, and insects, and enhancing growth and competitiveness of the host [4].

The roots of *Scutellaria baicalensis*, also called Scutellariae Radix, are an important herbal medicine in Traditional Chinese Medicine to treat diarrhea, dysentery, allergy, inflammation, jaundice, hypertension, pyrexia, hepatitis, and respiratory infections [5,6]. It is commonly known as skullcap, huang-qin (China), hwang-geum (Korea) and ogon or wogon (Japan) and is listed in Chinese, Korean, and Japanese Pharmacopoeias [7,8]. Interestingly, the inner part of young roots rot or decay when it gets older. There is very limited information on endophytes associated with the roots of *Scutellaria baicalensis* and its decaying process. In this study, a fungal strain of *Fusarium* sp. QF001 was isolated from freshly harvested old rotten roots, but any of fungal strain was not isolated from the freshly collected unrotten young roots of *Scutellaria baicalensis*.

*Fusarium* is a cosmopolitan genus of filamentous ascomycete fungi, which is widely distributed in plants and soils worldwide due to its ability to grow on wide range of substrates with efficient mechanisms of spore dispersal [9,10]. Thus, different species of *Fusarium* are prevalent in different environments such as temperate zone, desert, alpine, and arctic areas and in fertile cultivated land. As *Fusarium* is a soil-born fungi, eventually they get associated with plant, as either parasites or saprophytes [9].

Previously, *Fusarium* sp. were considered as plant pathogens that cause diseases like crown rot, head blight and scab on cereal grains, vascular wilts on a wide range of horticultural crops, root rots, cankers, and other diseases such as pokkah-boeng on sugarcane and bakanae disease of rice, with some producing mycotoxins (such as fumonisins, zearalenones, and trichothecenes) on plants [11]. Later, *Fusarium solani* was identified as an endophyte and reported to produce a diversity of outstanding bioactive secondary metabolites such as javanicin, fusarubin, anhydro-fusarubin, solaniol, marticin, and nectraiafurone [12]. This class of compounds is of interest due to the broad spectrum of their biological activities, such as antibacterial, antifungal, phytotoxic, insecticidal, and cytotoxic properties [12].

In continuation of discovery of anti-inflammatory secondary metabolites from an endophytic fungi, the *Fusarium* sp. QF001, isolated from the roots of *Scutellaria baicalensis*, was investigated as part of our research program. In this study, chemical investigation of ethyl acetate extract led to the isolation of two new isomers of lucilactaene (compounds **2**–**3**) along with seven known compounds identified as fusarubin (**1**), solaniol (**4**), javanicin (**5**), lucilactaene (**6**), 9-desmethylherbarine (**7**), NG391 (**8**) and NG393 (**9**), as shown in Figure 1. The known metabolites were identified by comparison of their spectroscopic data with those reported in the literature. Here, we report the isolation and structure elucidation of new isomers of lucilactaene and the anti-inflammatory activities of the isolates **1**–**9** against LPS-induced inflammation in macrophage cells (RAW 264.7).

## 2. Results and Discussions

### 2.1. Isolation and Identification of Fusarium

The inner rotten roots of *Scutellaria baicalensis* were investigated for the presence of endophytic fungi. The fungus QF001 with characteristic red pigmentation morphology was found on potato dextrose agar (PDA) plates. Based on internal transcribed spacer (ITS) sequencing and morphology, the endophytic fungus QF001 was identified as a *Fusarium* sp. (blast top stain: *Fusarium* sp. KC-2010ba strain USMFSSC10, 91.67% of similarity). Since, *Fusarium* sp. QF001 was found only from the inner rotten part of old roots of *Scutellaria baicalensis* and it suggested that the fungal strain is responsible for the root rotting of *S. baicalensis*. Furthermore, it could change the secondary metabolites in an herbal medicine Scutellariae Radix.

### 2.2. Isolation of Secondary Metabolites from the Fermentation Broth of Fusarium sp. QF001

The crude extracts were fractionated into seven fractions (A-G) by normal phase (NP) vacuum liquid chromatography (VLC) with stepped gradient elution (MC:MeOH = 100:0, 100:1, 50:1, 20:1, 10:1, 1:1, 0:100). The fraction C (eluting with MC:MeOH = 50:1) was subjected to semi-preparative reversed phase (RP) HPLC to give six peak-collected samples (C1-C6, Appendix A). Among them, only three isolates [C1 (Rt 19 min), C4 (Rt 25 min), and C5 (Rt 33 min)] were found to be relatively stable and their structures were determined to be fusarubin (**1**), solaniol (**4**), and javanicin (**5**) by comparing measured spectroscopic data with the reported records [13,14,15,16].

The other three peak-collected samples (C2 (Rt 22 min), C3 (Rt 23.5 min), and C6 (Rt 38 min)) were considered to be a mixture based on the ^1^H-NMR spectra (data not shown). Particularly, the two major samples C2 and C6 were analyzed by reinjection onto a semi-preparative RP HPLC column, and both samples showed similar mixture chromatographic profiles (Appendix A). Additionally, the three major peaks (C6a-C6c) of the analytical RP HPLC chromatogram of C6 were further purified and they showed almost similar chromatographic profiles with the parent sample (C6) on analytical RP HPLC (Appendix A). These results were also found with C2 and its three major peaks (C2a-C2c) (Appendix A).

To check the chemical conversion among the three major compounds of C6 (C6a-C6c), the three compounds were separately collected from the analytical RP HPLC. Each collected sample (C6a-C6c) was stored in the mobile phase (31% aqueous acetonitrile) of the isolation, and the chromatographic profile of each sample was analyzed on RP HPLC at 1 h and 24 h after peak collection (Appendix A). In the case of the C6a sample, the peak intensity of C6a decreased and that of C6c increased after 24 h of isolation. Whereas, in case of the C6c sample, the peak intensity of C6a increased following a decrease in C6c peak intensity. These results suggest that C6a-C6c are converted into one another and they are in the equilibrium state under the protic solvent conditions used in RP HPLC (mixture of H_2_O and acetonitrile).

The chemical stability of C6a-C6c was also tested under non-polar aprotic solvents (1-hexane and ethyl acetate) frequently used in NP HPLC. Three isolated compounds (C6a, C6b, and C6c) were isolated from NP HPLC and they were stored in the mobile phase of isolation (40% 1-hexane in ethyl acetate) for 24 h after isolation. In analytical NP HPLC, the isolated compounds C6a-C6c were observed to be relatively stable for 24 h after purification (Appendix A). It shows that solvents also play important role for the stability of compounds. Normally, RP chromatography is a common choice for isolation where isolated compounds are exposed to a polar protic solvent. Whereas, in case of NP chromatography isolated compounds are exposed to a non-polar aprotic solvent. The preference for NP chromatography over RP chromatography is advantageous, not only for the stability of compounds but it also helps in evaporation in lower temperature. This minimizes the exposure of light sensitive compounds to light which ultimately contributes in stability of compounds. Therefore, further isolation of relatively unstable compounds from fractions C and D (elution with MC:MeOH=20:1, 87.9 mg) were performed with NP HPLC.

Three major compounds in C2 and C6 [C2a (identical to C6a, **2**), C2b (identical to C6b, **3**) and C2c (identical to C6c, **6**)] along with **7** were successfully purified. In addition, NP HPLC on fraction E (elution with MC:MeOH = 10:1, 210.6 mg) led to the isolation of **8** and **9**. Based on spectroscopic analyses, including UV spectra, 1D and 2D NMR spectra, and MS spectra, compounds **6**–**9** were identified as lucilactaene (**6**), 9-desmethylherbarine (**7**), NG391 (**8**) and NG393 (**9**) [13,17,18].

### 2.3. Planar Structure Elucidation of ***2*** and ***3***

Compound **2** was isolated as a yellow amorphous solid with molecular formula C_22_H_27_NO_6_ as determined by a protonated ion peak at *m/z* 402.1921 [M + H]^+^ in high resolution mass spectrum. The spectroscopic data, including ^1^H- and ^13^C-NMR spectra, as well as MS and UV spectra of **2** were very similar to those of lucilactaene (**6**), which contained pentaene and furanopyrrolidone moiety (Figure 1, Table 1). The interpretation of 1D and 2D NMR data of **2** confirmed the presence of pentaene and furanopyrrolidone moieties (Figure 2). However, several ^1^H-NMR peaks corresponding to the pentaene parts of **2** were observed to be shifted (three upfield shifted olefinic signals: H-8 at δ_H_ 6.58 (1H, t, *J* = 11.4), H-9 at δ_H_ 6.40 (1H, t, *J* = 11.4), and H-10 at δ_H_ 7.97 (1H, d, *J* = 11.4); a downfield shifted olefinic signal: H-7 at δ_H_ 6.95 (1H, dd, *J* = 15.0, 11.4)). In particular, the coupling constant between H-8 and H-9 in **2** was measured as 11.4 Hz, whereas that of **6** was observed as 14.5 Hz. This indicated the 8*Z*-configuration of **2**, and this was further confirmed by a key nuclear Overhauser effect spectroscopy (NOESY) correlation between H-7 and H-10. The NOESY correlations (H-1 and H-4; H-4 and H-6; H-6 and H-8; H-7 and H-22; H-9 and H-23) and relatively large coupling constants between H-6 and H-7 (*J* = 15.0) in ^1^H-NMR spectrum suggested *E* geometry on the conjugated double bonds except for 8*Z* (Figure 2).

Heteronuclear multiple bond correlation (HMBC) crosspeaks from δ_H_ 4.29 (1H, d, H-14) to δ_C_ 197.3 (C-12), 57.0 (C-13), 170.4 (C-17), and 68.7 (C-19); from δ_H_ 4.38 (1H, br s, H-13) to δ_C_ 197.3 (C-12), 85.8 (C-14), 94.5 (C-15) and 170.4 (C-17) suggested the presence of a furanopyrrolidone found in lucilactaene (**6**). Comprehensive analysis of the correlation spectroscopy (COSY), HSQC, and HMBC data affirmed the planar structure of **2**, which was the 8*Z*-geometric isomer of lucilactaene (Figure 2).

Compound **3** was also obtained as a yellow amorphous solid with molecular formula C_22_H_27_NO_6_ based on a protonated ion peak at *m/z* 402.1917 [M + H]^+^ in its high resolution mass spectrum. The similarity in the *m/z* value of the protonated molecule and the NMR spectra of **3**, **2,** and **6** suggested that **3** belongs to the lucilactaene structure class. Although the NMR spectra of **6** were similar to those of lucilactaene, there was a markedly upfield shifted signal of H-4 at δ_H_ 6.10 (1H, s) and downfield shifted methyl signal of H-22 at δ_H_ 2.03 (1H, d, *J* = 1.3). In addition, the chemical shift of H-1 at δ_H_ 1.72 (1H, dd, *J* = 7.2, 1.4) and H-2 at δ_H_ 7.06 (1H, qd, *J* = 7.2, 1.0) indicated that **3** is the 4*Z*-isomer of lucilactaene, which was also strongly supported by a NOESY correlation between H-4 and H-22 (Figure 3). All the other olefinic configurations (except 4*Z*) were assigned as *E* type, which was supported by relatively large coupling constants (^3^*J*_6-7_ = 15.3 and ^3^*J*_8-9_ = 14.6) and several key NOESY correlations (H-22 and H7; H-9 and H-7/H-23; H-8 and H-6/H-10).

### 2.4. Configuration Analysis of ***2***, ***3***, ***4***, and ***6***

Three of the furanopyrrolidone-containing compounds (**2**, **3**, and **6**) showed key NOSEY correlations between H-14 and 15-OH (Figure 2 and Figure 3). In addition, H-13 and H-14, two methinyl protons were observed as singlets, broad singlet, or doublet, with the coupling constants less than 1.0 Hz (Table 1). These two pieces of evidence support the relative configurations of furanopyrrolidones in **2**, **3**, and **6**. These observations were compared to the energy-minimized structures of furanopyrrolidone with four possible stereoisomers (Figure 4 and Appendix A). Case I (13*S**, 14*S**, and 15*S**) and case III (13*S**, 14*R**, and 15*R**) were calculated to have dihedral angles between H-14 and 15-OH of 172.7° and 178.4°, respectively, which are inconsistent with the observed NOESY correlations between H-14 and 15-OH. Furthermore, the small vicinal coupling between H-13 and H-14 in **2**, **3,** and **6** indicated that the dihedral angle between these two protons is close to 90°, which is well matched to the case IV (13*S**, 14*R**, and 15*S**; dihedral angle between H-13 and H-14: 96.4°). Previously, naturally occurring lucilactaene was reported to possess 13*S*, 14*R*, and 15*S* configurations with the positive specific rotation ([α]_D_^25^ + 36.6). The specific rotations of **2**, **3**, and **6** were measured as +23, +17, and +31, respectively. Therefore, the absolute configurations of furanopyrrolidones in **2**, **3**, and **6** could be speculated to be 13*S*, 14*R*, and 15*S*.

As aforementioned, **4** was identified as solaniol. Previously, (+)-solaniol was reported as a natural product [16,19]. The measurement of the specific rotation of **4** in this study was +25 which indicated that **4** is (+)-solaniol. However, its absolute configuration has never been assigned before. Its absolute configuration was deduced by comparing measured specific rotation with calculated value (Appendix A). Computer-assisted calculation of specific rotation of *R*-solaniol resulted in the positive sign of specific rotation and the sign of *S*-solaniol was calculated to be opposite, indicating **4** is (+)-*R*-solaniol.

### 2.5. Biological Activities of Isolated Compounds

Compounds **1**, **4**, **5**, and **6** were previously reported to be cell cycle inhibitors or cytotoxins against cancer cells. Therefore, the cell viability of the isolated compounds were first tested on the macrophage (RAW264.7) and none of the isolates show cytotoxicity at concentrations ranging between 0.7 to 50 μM (Appendix A, Appendix A). Thus, the biological activity of isolated compounds and dexamethasone (DEX; control drug) was evaluated at the concentration of 10 μM. Inflammatory disease is activated by relative oxidative stress and pro-inflammatory cytokines interlukin-6 (IL-6) and tumor associated factor-α (TNF-α). Therefore, the anti-inflammatory activity of isolated compounds via production of NO (nitric oxide), TNF-α, and IL-6 in LPS-induced macrophage cells was evaluated (Figure 5). Lipopolysaccharides (LPS) notably increased NO production in macrophage cells. However, treatment of DEX or isolated **1**–**9** inhibited NO production. Similarly, the expression of pro-inflammatory cytokines such as TNF-α and IL-6 was suppressed after treatment of DEX or **1**–**9** on LPS-induced macrophage cells (Figure 5).

Compound **1** was previously reported as a phytotoxin, cytotoxin against human cancer cells, a anti-parasitic agent against *Trichomonas vaginalis*, and an anti-tuberculosis agent [20,21,22,23]. Compound **4** was known with the hepatic glucose production inhibiting activity and cytotoxicity [24]. Compound **5** was also reported to have cytotoxicity and antibacterial activity as well as anti-parasitic activity and hepatic glucose production inhibiting activity [22,25,26]. Lucilactaene (**6**) was well known as cell cycle inhibitor [17]. Compounds **6**, **8** and **9** were reported to be alkaline phosphatase inhibitors in bone morphogenetic protein-stimulated myoblastoma cells [27]. However, it has never been investigated on the anti-inflammatory activity with inhibiting NO production and pro-inflammatory cytokine expression (TNF-α and IL-6).

## 3. Materials and Methods

### 3.1. Materials

NMR spectra were acquired with 250 MHz (Bruker, Madison, WI, USA) and 600 MHz (Varian, Palo Alto, CA, USA) spectrometers. Low resolution electron spray ionization Mass (LRESIMS) spectra was obtained using an Agilent 1260 Infinity LC along with quadrupole 6120 ESI mass spectrometer (Agilent Technologies, Santa Clara, CA, USA). A high resolution mass spectrometer (FAB negative mode) was used to acquire mass data for new isomers. Column chromatography was performed using C18 reversed column and silica gel. HPLC purifications were carried out on a Phenomenex Luna C18 column or Hector C18 or Intersil Sil column (250 × 4.6 mm; 250 × 10 mm) with Waters 616 HPLC system (Waters, Milford, MA, USA) equipped with a photodiode array (PDA) 996 detector. All the chemicals used were of analytical or chromatographic grade.

### 3.2. Fungus Isolation and Identification

The endophytic fungus was isolated from healthy fresh old roots of *Scutellaria baicalensis* collected from Medicinal Herbal Garden of Daegu Catholic University (35°54′43.41″ N 128°48′13.94″ E, South Korea) in November 2016. To isolate the endophytic fungi, the collected material was cut into small pieces and thoroughly washed in sterile water. The washed *Scutellaria baicalensis* piece was surface-disinfected by soaking in 95% ethanol for 1 min and in bleach for 3 min, and it was rinsed subsequently with sterile distilled water for three times. Then it was transferred to PDA plates such that the cut edges were in direct contact with agar media. After several days of incubation, the fungal strains grown were further purified several times to obtain pure fungal strain. The endophytic fungus was identified as *Fusarium* sp. using combination of morphological characteristics and ITS gene sequences (blast top strain: *Fusarium* sp. KC-2010ba strain USMFSSC10 with 91.67% similarity).

### 3.3. Extraction and Isolation of ***1*–*****4***

The fungus was grown in 4 L (1 L/flask × 4) of potato dextrose broth medium in distilled water for 14 days at 37 °C under static condition. After completion of incubation, culture broth was extracted three times with ethyl acetate. The combined organic layers were dried over anhydrous Na_2_SO_4_ and concentrated under reduced pressure with rotary evaporator to obtain dark red color solid crude extracts (558.0 mg). The crude extracts were fractionated into seven fractions (A-G) by NP VLC with stepped gradient elution (MC:MeOH = 100:0, 100:1, 50:1, 20:1, 10:1, 1:1, 0:100). The fractions C (eluted with MC:MeOH = 50:1, 98.9 mg) and D (eluted with MC:MeOH = 20:1, 87.9 mg) were subjected to semi-preparative RP HPLC (Phenomenex Luna C18, 250 × 10 mm, 5 μm, 2.0 mL/min, mobile phase: A (0.05% formic acid in H_2_O), B (0.05% formic acid in acetonitrile), gradient elution: B 44% (0 to 45 min), B 44 to 100% (45 to 50 min), B 100% (50 to 60 min)) to give purified compounds **1**–**4** and two peak-collected samples (C2 and C6).

### 3.4. Time Dependent Analysis on Conversion of C2 and C6

Two peak-collected samples showed almost identical HPLC profiles on analytical RP HPLC. The isomerization of the isolated furanopyrazolidine compounds (C2a-C2c, and C6a-C6c) was monitored by analytical HPLC on both NP and RP stationary phases. RP HPLC was performed with analytical RP HPLC (Phenomenex Kinetex C18 column, 250 × 4.6 mm, 5 μm, 0.8 mL/min, isocratic elution (H_2_O:acetonitrile = 69:31), PDA monitoring at 210 and 365 nm). NP HPLC was performed on analytical NP HPLC (Inertsil Sil column, 250 × 4.6 mm, 5 μm, 0.7 mL/min, 1-hexane:ethyl acetate = 40:60), PDA monitoring at 210 and 365 nm).

### 3.5. Isolation Strategy and Purification of Photosensitive Polyenes ***5***–***9***

Two peak-collected samples (C2 and C6) from fractions C and D were subjected to NP HPLC to give pure **5**–**7** (Hector Sil column, 250 × 10 mm, 5 μm, 1.8 mL/min, gradient elution: 1-hexane:ethyl acetate= 40:60 (isocratic for 35 min), 40:60 to 35:65 (gradient for 17 min), PDA detection at 260 and 365 nm). To minimize the isomerization by light and high temperature, experiments were performed under dim light with use of amber-colored vials for collecting isolated compounds. After that to impede exposure to high temperature, the vial consisting isolates were kept on ice, and the solvent was evaporated under nitrogen gas flow only. NP HPLC [Hector Sil column, 250 × 10 mm, 5 μm, 2.0 mL/min, gradient elution: 1-hexane:ethyl acetate= 1 5:85 (isocratic for 50 min), PDA detection at 254 and 365 nm] on fraction E resulted in the isolation of **8** and **9**.

### 3.6. Characterization of Isolated Compounds

*Fusarubin* (**1**), Red amorphous solid; ^1^H-NMR (250 MHz, CDCl_3_) δ 12.94 (s, 1H), 12.67 (s, 1H), 6.18 (s, 1H), 4.89 (s, 2H), 3.94 (s, 3H), 3.04 (d, *J* = 18.2 Hz, 1H), 2.72 (dd, *J* = 18.2, 1.4 Hz, 1H), 2.28 (br s, 1H), 1.66 (s, 3H). ^13^C-NMR (63 MHz, CDCl_3_) δ 184.9, 178.4, 160.9, 160.8, 157.4, 137.3, 109.8, 107.7, 94.4, 58.7, 56.9, 32.3, 29.5; LRESIMS [M + Na]^+^
*m/z* 329.1.

*8Z-Isomer of lucilactaene* (**2**), Yellow amorphous solid; [α]_D_^25^ + 23 (c 0.1, CHCl_3_); ^1^H-NMR (600 MHz, CDCl_3_) and ^13^C-NMR (150 MHz, CDCl_3_) see Table 1; HRESIMS [M + H]^+^
*m/z* 402.1921 (calcd for C_22_H_28_NO_6_^+^, 402.1911).

*4Z-Isomer of lucilactaene* (**3**), Yellow amorphous solid; [α]_D_^25^ + 17 (c 0.1, CHCl_3_); ^1^H-NMR (600 MHz, CDCl_3_) and ^13^C-NMR (150 MHz, CDCl_3_) see Table 1; HRESIMS [M + H]^+^
*m/z* 402.1917 (calcd for C_22_H_28_NO_6_^+^, 402.1911).

*Solaniol* (**4**), Red amorphous solid; [α]_D_^25^ + 25 (c 0.1, CH_3_OH); ^1^H-NMR (250 MHz, CDCl_3_)) δ 13.34 (s, 1H), 13.07 (s, 1H), 6.21 (s, 1H), 4.21–4.05 (m, 1H), 3.94 (s, 1H), 2.97–2.93 (m, 1H), 2.35 (s, 1H), 1.79 (br s, 1H), 1.57 (s, 3H), 1.32 (d, *J* = 6.2 Hz, 1H). ^13^C-NMR (CDCl_3_) δ 203.7, 184.5, 178.0, 160.7, 160.3, 160.2, 142.4, 134.2, 109.8, 56.8, 41.3, 30.0, 12.9; LRESIMS [M + H]^+^
*m/z* 293.1.

*Javanicin* (**5**), Red amorphous solid; ^1^H-NMR (250 MHz, CDCl_3_) δ 13.26 (s, 1H), 12.86 (s, 1H), 6.21 (s, 1H), 3.94 (s, 3H), 3.91 (s, 2H), 2.30 (s, 3H), 2.24 (s, 3H). ^13^C-NMR (63 MHz, CDCl_3_) δ 203.9, 184.5, 177.9, 161.6, 160.8, 160.5, 142.6, 134.3, 109.8, 109.8, 108.6, 56.9, 41.4, 30.1, 13.0; LRESIMS [M + H]^+^
*m/z* 291.0.

*Lucilactaene* (**6**), Yellow amorphous solid; [α]_D_^25^ + 31 (c 0.1, CHCl_3_); ^1^H-NMR (600 MHz, CDCl_3_) and ^13^C-NMR (150 MHz, CDCl_3_) see Table 1; LRESIMS [M + H] ^+^
*m/z* 402.1.

*9-Desmethylherbarine* (**7**), Red amorphous solid; ^1^H-NMR (250 MHz, CDCl_3_) δ 12.12 (s, 1H), 7.19 (d, *J* = 2.5 Hz, 1H), 6.62 (d, *J* = 2.5 Hz, 1H), 4.73 (s, 2H), 3.90 (s, 3H), 2.87 (d, *J* = 19.0 Hz, 1H), 2.54 (d, *J* = 19.0 Hz, 1H), 2.31 (s, 1H), 1.62 (s, 4H); LRESIMS [M + Na]^+^
*m/z* 313.0.

*NG-391* (**8**), Yellow amorphous solid; [α]_D_^25^ + 22 (c 0.1, CHCl_3_); ^1^H-NMR (250 MHz, CDCl_3_ δ 7.48 (d, *J* = 11.4 Hz, 1H), 7.03 (s, 1H), 6.99 (qd, *J* = 7.2, 0.8 Hz, 1H), 6.79 (dd, *J* = 14.7, 10.3 Hz, 1H), 6.65 (dd, *J* = 14.7, 11.4 Hz, 1H), 6.62 (d, *J* = 15.4 Hz, 1H), 6.44 (dd, *J* = 15.4, 10.3 Hz, 1H), 6.21 (s, 1H), 5.11 (s, 1H), 4.09 (m, 1H), 4.03 (d, *J* = 1.1 Hz, 1H), 3.95 (m, 1H), 3.75 (s, 2H), 2.72 (br.s, 1H), 2.12 (m, 1H), 2.10 (m, 1H), 1.97 (s, 1H), 1.75 (dd, *J* = 7.2, 1.1 Hz, 2H), 1.71 (d, *J* = 0.6 Hz, 2H). ^13^C-NMR (63 MHz, CDCl_3_) δ 190.0, 169.9, 167.6, 145.3, 143.6, 142.3, 140.6, 138.2, 134.0, 130.6, 128.6, 128.1, 128.1, 85.5, 63.8, 61.9, 58.6, 52.1, 36.0, 16.0, 14.4, 11.5; LRESIMS [M + H]^+^
*m/z* 418.2.

*NG-393* (**9**), Yellow amorphous solid; [α]_D_^25^ + 16 (c 0.1, CHCl_3_); ^1^H-NMR (250 MHz, CDCl_3_ δ 7.89 (d, *J* = 11.4 Hz, 1H), 6.97 (m, 1H), 6.96 (m, 1H), 6.73 (m), 6.56 (d, *J* = 15.2, 1H), 6.52 (t, *J* = 11.1 Hz, 1H), 6.39 (t, *J* = 11.2 Hz, 1H), 6.22 (s, 1H), 5.30 (s, 1H), 4.09 (s, 1H), 4.06 (m, *J* = 10.9 Hz, 2H), 3.97 (m, 1H), 3.75 (s, 3H), 2.74 (br s, 1H), 2.14 (m), 2.12 (m), 6.5 Hz, 2H), 1.98 (s, 3H), 1.75 (dd, *J* = 7.2, 0.9 Hz, 5H), 1.71 (s, 4H). ^13^C-NMR (63 MHz, CDCl_3_) δ 190.0, 170.2, 168.0, 142.5, 140.8, 139.5, 138.8, 138.6, 134.5, 130.4, 127.3, 124.1, 124.1, 86.0, 64.0, 62.0, 58.9, 52.2, 35.8, 29.85, 16.1, 14.8, 11.3; LRESIMS [M + H]^+^
*m/z* 418.2.

### 3.7. Energy Minimization of Four Possible Stereoisomers of Furanopyrrolidones

The stereoisomers of furanopyrrolidones were subjected to conformational analysis using Spartan 16 (Wavefunction, Irvine, CA, USA) and Merck Molecular Force Field (MMFF). Each conformers of four stereoisomer having low energy with less than 10% of Boltzmann distribution were further analyzed for energy minimization by Gaussian 16 (Gaussian Inc., Wallingford, CT, USA) with B3LYP function and 6-31+G (d, p) basis set in chloroform. The 3D structure of lowest energy conformers of four stereoisomers were chosen for calculation of dihedral angles between H-14 and 15-OH.

### 3.8. Computer-Assisted Calculation of Specific Rotation of ***4***

A conformational study of *R*- and *S*- solaniol was performed by the Spartan 16 (Wavefunction,) using MMFF. The conformers with less than 10% of Boltzmann population were chosen for further calculation, respectively. The energy minimization of each conformers was carried out with B3LYP function and 6-311+G (2d, p) basis set in MeOH by Gaussian 16 (Gaussian Inc., Wallingford, CT, USA). The optical rotations of energy-minimized conformers were calculated in MeOH at B3LYP/6-311++G (2d,2p) level with 589 nm by Gaussian 16. The calculated optical rotations of *R*- and *S*- solaniol were averaged on the basis of the Boltzmann population of conformers, respectively.

### 3.9. Cell Culture and Cell Growth Analysis

Murine macrophage RAW264.7 cells were acquired from the Korea Cell Line Bank (Seoul, South Korea). All cells were cultured in Dulbecco’s Modified Eagle’s Medium supplemented with 10% fetal bovine serum and 1% penicillin-streptomycin, and incubated in a humidified incubator with 5% CO_2_ atmosphere at 37 °C.

### 3.10. Measurement of NO, TNF-α, and IL-6

RAW264.7 cells with 1 × 10^6^ cells/mL, at the logarithmic phase, were inoculated into 24-well plates overnight, and pretreated with 10 μM of isolated various compounds for 1 h followed by stimulation with 1 μg/mL LPS for additional 24 h, respectively. After stimulation, cell culture media were collected and centrifuged at 1200 rpm for 4 min at 4 °C. The supernatants were sampled for TNF-α, IL-6, and NO assays. TNF-α and IL-6 were quantitated using Mouse TNF-α and IL-6 ELISA kits following manufacturer’s protocols. NO was quantitated by detecting its stable oxidative metabolite, nitrite. Briefly, 50 μL culture media was blended with Griess reagent (Promega, Madison, WI, USA) of the same volume and incubated for 30 min at 37 °C. Absorbance was read at 540 nm using a microplate reader. The nitrite concentration was calculated by the calibration curve using sodium nitrate standard solutions.

### 3.11. Statistical Analysis

All data are expressed as mean ± standard deviation from at least three repeated experiments. Statistical significance was determined using an unpaired Student’s *t*-test with GraphPad Prism version 5 (GraphPad Software Inc., La Jolla, CA, USA) Differences with p-values of lower than 0.05 were considered statistically significant.

## 4. Conclusions

In this study, a chemical investigation on the culture broth of *Fusarium* sp. QF001 isolated from *Scutellaria baicalensis* led to the isolation of compounds **1**–**9**. Among the isolates, **2** and **3** were found to unstable in light conditions after RP chromatography. Stability monitoring by HPLC provided an insight and evidence for an equilibrium among **2**, **3**, and **6**. The structural conversion between **2**, **3**, and **6** were found to be inhibited under NP chromatography conditions with minimized exposure to light and polar protic solvents. Therefore, the preferred strategy of using NP chromatography over RP led to the successful isolation of the relatively unstable isomers of lucilactaene in a controlled environment and their structures were identified. The configurations of the compounds in lucilactaene class were assigned by analysis on ^1^H-^1^H coupling constants and NOESY correlations in combination with specific rotation comparison. In addition, (+)-solaniol (**4**) was analyzed to be *R*-solaniol based on computer-assisted calculation of specific rotation. Compounds **1**–**9** inhibited the production of NO in murine macrophages and downregulated expression of TNF-α and IL-6.

The results presented in this study show that NP chromatography with aprotic solvent conditions can be an appropriate isolation method for unstable polyene-containing compounds because *E*, *Z*-isomerization of polyene structures are solvent dependent. Also, the solvents frequently used in NP chromatography are more volatile than the solvents used in RP chromatography, and this will shorten the time of light exposure of compounds during the evaporation process. In the future, the effect of isomerization of lucilactaene on biological activities is worth exploring. In addition, the study of endophytic fungi and their host-plant interactions as well as effect of secondary metabolites produced by endophytic fungi on the pharmacological use of host plants could be a topic for future research.

## Figures and Tables

**Figure 1 molecules-25-00923-f001:**
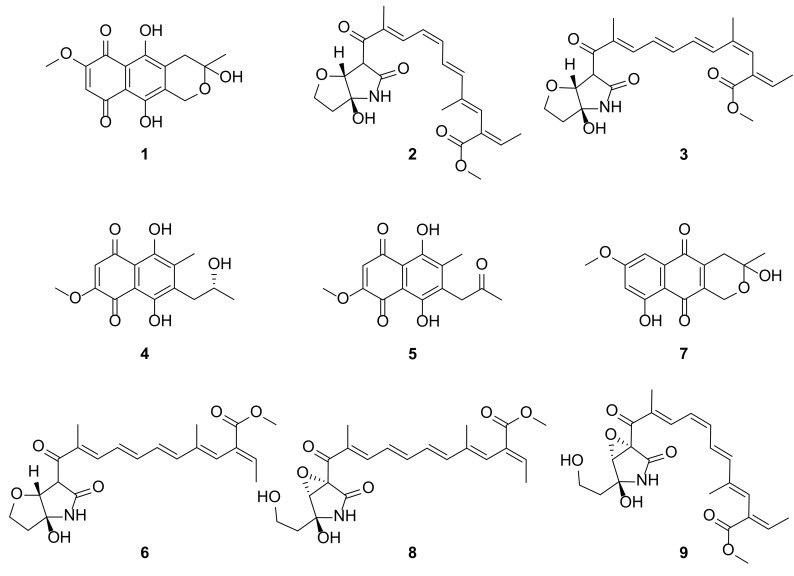
Chemical structures of isolated compounds: fusarubin (**1**), *8Z*-isomer of lucilactaene (**2**), *4Z*-isomer of lucilactaene (**3**), (+)-solaniol (**4**), javanicin (**5**), lucilactaene (**6**), 9-desmethylherbarine (**7**), NG-391 (**8**) and NG-393 (**9**).

**Figure 2 molecules-25-00923-f002:**
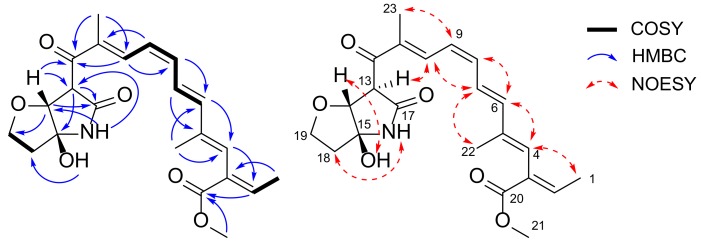
Key ^1^H-^1^H COSY, HMBC, and NOESY correlations of **2**.

**Figure 3 molecules-25-00923-f003:**
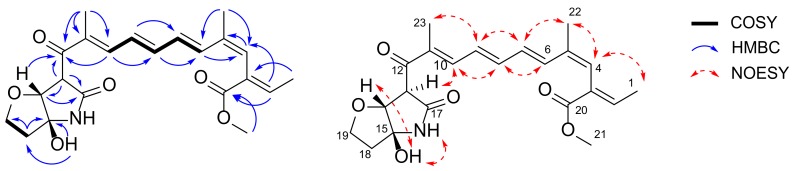
Key ^1^H-^1^H COSY, HMBC, and NOESY correlations of **3**.

**Figure 4 molecules-25-00923-f004:**
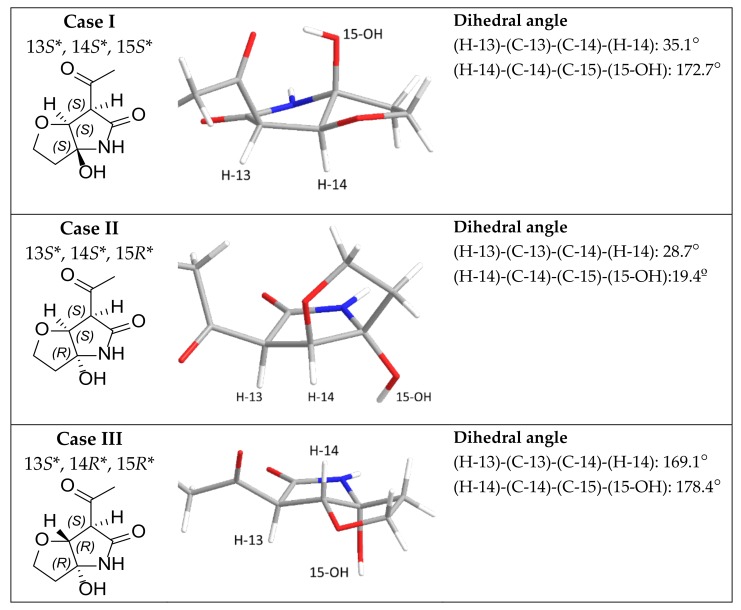
Energy-minimized structures of four possible stereoisomers of furanopyrrolidone moiety of **2**, **3**, and **6**.

**Figure 5 molecules-25-00923-f005:**
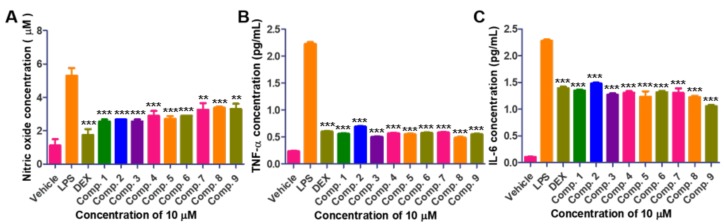
Effect of isolated **1**–**9** in macrophage cells. (**A**) Effect of **1**–**9** on LPS-induced NO production in macrophage cells (1 μg/mL LPS and combination of 10 μM dexamethasone (DEX), and isolated compounds). Effect of **1**–**9** on expression level of (**B**) TNF-α and (**C**) IL-6 in medium as determined by enzyme-linked immunosorbent assay (ELISA) in LPS-induced macrophage cells. Macrophage cells were pre-treated with indicated concentration of DEX, and isolated compounds for 1 h and then with LPS (1 μg/mL, 24 h). (** *p* < 0.001, *** *p* < 0.001 VS LPS).

**Table 1 molecules-25-00923-t001:** NMR spectroscopic data for **2**, **3** and **6** in CDCl_3_ (δ_H_ in ppm and *J* in Hz).

No.	Compound 2 ^a,b^		Compound 3 ^a,b^		Compound 6 ^c^	
	δ_H_ (Multiplicity, *J* Hz)	δ_C_, Type	δ_H_ (Multiplicity, *J* Hz)	δ_C_, Type	δ_H_ (Multiplicity, *J* Hz)	δ_C_, Type
1	1.75 (dd, 7.2, 1.3)	16.2, CH_3_	1.72 (dd, 7.2, 1.4)	16.1, CH_3_	1.75 (dd, 7.2, 1.2)	16.1, CH_3_
2	7.00 (qd, 7.2, 0.6)	140.8, CH	7.06 (qd, 7.2, 1.0)	141.2, CH	7.00 (q, 7.2)	140.7, CH
3		130.4, C		130.0, C		130.5, C
4	6.24 (br s)	128.3, CH	6.10 (br s)	126.5, CH	6.24 (s)	128.3, CH
5		138.4, C		136.5, C		138.1, C
6	6.64 (d, 15.1)	143.0, CH	6.45 (d, 15.3)	136.8, CH	6.65 (d, 15.0)	142.5, CH
7	6.95 (dd, 15.1, 11.7)	123.6, CH	6.51 (dd, 15.3, 10.4)	129.9, CH	6.47 (dd 15.0, 10.6)	128.5, CH
8	6.58 (t, 11.4)	139.8, CH	6.78 (dd, 14.6, 10.4)	143.9, CH	6.85 (dd, 14.6, 10.6)	143.8, CH
9	6.40 (t, 11.4)	124.1, CH	6.68 (dd, 14.6, 11.5)	128.6, CH	6.67 (dd, 14.6, 11.0)	128.2, CH
10	7.97 (d, 11.6)	139.8, CH	7.44 (d, 11.5)	145.5, CH	7.48 (d, 11.0)	145.7, CH
11		134.8, C		134.6, C		134.3, C
12		197.4, C		197.1, C		197.2, C
13	4.38 (br s)	57.3, CH	4.33 (br s)	56.8, CH	4.36 (br s)	56.8, CH
14	4.29 (d, 0.6)	85.8, CH	4.25 (d, 0.7)	85.8, CH	4.25 (s)	85.9, CH
15		94.5, C		94.6, C		94.7, C
15-OH	5.00 (s)		4.97 (s)		5.04 (s)	
16-NH	6.18 (br s)		6.06 (br s)		6.11 (s)	
17		170.4, C		170.6, C		170.9, C
18a	2.28 (ddd, 12.7, 6.5, 3.6)	37.6, CH_2_	2.27 (m)	37.6, CH_2_	2.28 (m)	37.6, CH_2_
18b	2.45 (dt, 12.7, 8.8)		2.43 (dt, 12.7, 8.8)		2.43 (m)	
19a	4.05 (td, 8.8, 6.5)	68.7, CH_2_	4.03 (td, 8.8, 6.4)	68.7, CH_2_	4.02 (td, 8.8, 6.5)	68.7, CH_2_
19b	4.14 (td, 8.8, 3.6)		4.13 (td, 8.8, 3.8)		4.13 (td, 8.8, 3.3)	
20		167.7, C		167.6, C		167.6, C
21	3.75 (s)	52.1, CH_3_	3.76 (s)	52.1, CH_3_	3.75 (s)	52.1, CH_3_
22	1.79 (d, 1.0)	14.7, CH_3_	2.03 (d, 1.3)	20.0, CH_3_	1.72 (d, 1.0)	14.4, CH_3_
23	1.96 (d, 0.5)	11.5, CH_3_	1.95 (d, 0.8)	11.7, CH_3_	1.95 (s)	11.7, CH_3_

^a^ Assignments based on HMQC, HMBC and ^1^H-^1^H COSY experiments. ^b^ 600 MHz for ^1^H-NMR and 150 MHz for ^13^C-NMR in CDCl_3_. ^c^ 250 MHz for ^1^H-NMR and 63 MHz for ^13^C-NMR.

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
