# Peer review of "Isolation of Unstable Isomers of Lucilactaene and Evaluation of Anti-Inflammatory Activity of Secondary Metabolites Produced by the Endophytic Fungus Fusarium sp. QF001 from the Roots of Scutellaria baicalensis"

_molecules, 2020, doi:10.3390/molecules25040923_

Round 1
Reviewer 1 Report
The authors investigated the compounds produced by fungus from S. baicalensis, and the structure elucidations of the compounds were correctly carried out. Although the anti-inflammatory activity of the compounds were not so significant, this manuscript will be suitable for publication. Before acceptance, please check the following subjects:
As trans-cis isomerization might occur between extraction and starting separation, the authors should check the existence of compounds 2, 3, and 6 in the medium, in which the fungus grown. The authors should show that compounds 2, 3, and 6 were not the artifact(s), or they should mention the possibility that the isomerization might occur artificially during extraction and/or isolation. No data of compound 7 were written, in spite that the others were shown. Please show the data of 7.Author Response
As trans-cis isomerization might occur between extraction and starting separation, the authors should check the existence of compounds 2, 3, and 6 in the medium, in which the fungus grown. The authors should show that compounds 2, 3, and 6 were not the artifact(s), or they should mention the possibility that the isomerization might occur artificially during extraction and/or isolation.
: Thank you for the suggestions and comments. The extracts of Fusarium sp. QF001 was prepared by using EtOAc extraction from the microbial culture broth. The resulting extracts were subjected to NP VLC and the fractions were evaporated as fast as possible. Therefore, we think the exposure of the extracts and fractions to lights and protic solvents were minimized. The first attempt to isolate compounds 2, 3, and 6 from a portion of fraction C was done with RP HPLC, but we found isomerization between compounds 2, 3, and 6. We think we consumed relatively long time in the removal of solvents from the collected compounds and they were isomerized during the removal of solvents. Thus, the residual fraction C was subjected to NP HPLC for the isolation of compounds 2, 3, and 6. They were observed as peaks on NP HPLC and it resulted in the successful isolation of compounds without isomerization. Therefore, we think compounds 2, 3, and 6 were natural products rather than artefacts.
No data of compound 7 were written, in spite that the others were shown. Please show the data of 7.
: As described in lines 128-129 of revised manuscript, compound 7 was obtained from C2 and C6 along with compounds 2, 3, and 6 by NP HPLC. Its structure was confirmed as 9-desmethylherbarine based on UV spectra, 1H NMR spectra and MS spectra. In order to provide solid evidence of compound identity, we tried to acquire 13C NMR spectra and the other 2D NMR spectra. However, these efforts were not successful, due to the limited amount of compound 7. The 1H NMR chemical shifts data of 7 were included in the lines 299-301 of the revised manuscript and The 1H NMR spectra was provided as Figure S31 in the Supplementary Materials.
Reviewer 2 Report
These manuscript is brilliant, very interesting and current. The introduction is very well organized and fits perfectly into the theme under study. The methodology is also well described and easily understandable, although they use many different methodologies.
The results and discussion are well described and compared with similar studies performed by other authors. The tables, figures and graphics are very well formatted and organized which facilitates the reading and understanding of the results.
After these considerations, the article can be accepted in the present form.
Author Response
These manuscript is brilliant, very interesting and current. The introduction is very well organized and fits perfectly into the theme under study. The methodology is also well described and easily understandable, although they use many different methodologies. The results and discussion are well described and compared with similar studies performed by other authors. The tables, figures and graphics are very well formatted and organized which facilitates the reading and understanding of the results. After these considerations, the article can be accepted in the present form.
: Thank you for your positive review on our manuscript.
Reviewer 3 Report
The paper by Sailesh Maharjan et al described the purification and identification of 9 molecules from the endophytic fungus Fusarium sp. 7 of theses molecules were known molecules, and the two others are isoforms of lucilactaene. The authors show that the 9 molecule have a potential as anti-inflammatory drugs.
To my point of view, experiments are well conduct and sound.
I noted however that it seems that Figure 1 contain errors. The 4-Z and 8-Z isomers are inversed. The structure of the compound 3 is not good.
Regarding the biological activity of the 9 molecules, I have some question :
Lucilactaene seems to have cytotoxic effects, be a cell cycle inhibitor and could induced apoptosis at concentrations up to 5-10 µM.
You described no cytotoxic activity on your macrophage cells. However, have you tested others biological effects and, if so, what were the results?
In the same way, have you test your molecules on others biological material?
Because, if you conclude to the potential of these molecules as anti-inflammatory drugs, the multiple effects of these compounds could be a proble with potential secondary effects.
Author Response
To my point of view, experiments are well conduct and sound.I noted however that it seems that Figure 1 contain errors. The 4-Z and 8-Z isomers are inversed. The structure of the compound 3 is not good.
: All the authors deeply appreciate the comments of Reviewer 3. As pointed out, the erroneous names of compounds 2 and 3 in Abstract (line 27) and Figure 1 legend (line 44) were corrected. In addition, the structure of compound 3 in Figure 1 was re-drawn.
Regarding the biological activity of the 9 molecules, I have some question :
Lucilactaene seems to have cytotoxic effects, be a cell cycle inhibitor and could induced apoptosis at concentrations up to 5-10 μM. You described no cytotoxic activity on your macrophage cells. However, have you tested others biological effects and, if so, what were the results?
: Thank you for valuable comments. As reviewer suggested, we have conducted additional experiments to confirm no cytotoxic effect of the isolated compounds for macrophage cells. These results added in Results and Discussion (lines 199-202) and in the description on Supplementary Materials (line 377) as well as supplementary materials (Figure S36).
In the same way, have you test your molecules on others biological material?
Because, if you conclude to the potential of these molecules as anti-inflammatory drugs, the multiple effects of these compounds could be a proble with potential secondary effects.
Try to discuss the anti-inflammatory findings with a possible bioactive compounds mode of action against the tested cell lines.
: Thank you for your positive review on our manuscript. As suggested by the reviewer, we have included the detail of the multiple effect of compounds 1-9 in the revised manuscript (lines 218-226) with additional references (reference numbers 20-27).
Reviewer 4 Report
Dear Author, I reviewed the manuscript (molecules-700923) entitled Isolation of Unstable Isomers of Lucilactaene and Evaluation of Anti-inflammatory Activity of Secondary Metabolites Produced by an Endophytic Fungus Fusarium sp. QF001 from the roots of Scutellariae baicalensis. This manuscript presents relevant information about the use of fungus isolates with anti-inflammatory properties. However, the results and discussion of the obtained data can be improved. For this reason, I considered that this manuscript needs minor changes for being considered for its publication in this journal.
Additional comments.
Highlight the advantages of using these fungus isolates and their bioactivity.
Highlight the importance of this extraction and elucidate protocol with other habitual protocols.
Try to discuss the anti-inflammatory findings with a possible bioactive compounds mode of action against the tested cell lines.
Include future trends to keep working with the obtained data.
Include a statistical description in the figures that required it.
Try to conclude with a general statement of the most relevant part of this study.
Author Response
Dear Author, I reviewed the manuscript (molecules-700923) entitled Isolation of Unstable Isomers of Lucilactaene and Evaluation of Anti-inflammatory Activity of Secondary Metabolites Produced by an Endophytic Fungus Fusarium sp. QF001 from the roots of Scutellariae baicalensis. This manuscript presents relevant information about the use of fungus isolates with anti-inflammatory properties. However, the results and discussion of the obtained data can be improved. For this reason, I considered that this manuscript needs minor changes for being considered for its publication in this journal.
: Thank you for the valuable comments provided by Reviewer 4. We tried to describe our results little more detail and add several sentences on the discussion. Lines 86-89, 118-124, 199-202, 218-226 and 367-374 along with references 20-27 were added in the revised manuscript.
Additional comments.
Highlight the advantages of using these fungus isolates and their bioactivity.
: Fusarium sp. QF001 was isolated from the rotten parts of the roots of old medicinal plant Scutellaria bicalensis. Previously, flavonoids such as baicalin, baicalein and wogonin were reported as bioactive component of S. baicalensis. However, the endophytic fungus Fusarium sp. QF001-induced decaying can modify the metabolites in the Scutellaria bicalensis and this can result in the use of this herbal medicines. In the future, study of endophytic fungus and host-plant interaction as well as effect of secondary metabolites produced by endophytic fungus on pharmacological use of host plant can be scope for future research. These were emphasized and added in the manuscript lines 86-89 and 371-374.
Highlight the importance of this extraction and elucidate protocol with other habitual protocols.
: Recent years, most of isolation was achieved by organic solvent extraction followed by RP HPLC. However, the exposure of natural products to the protic solvents and the lights can induce the artificial changes on the natural products. In particular, the compound with polyene structure is easily isomerized under protic solvent and ambient light. Therefore, the compound potentially including polyenes need to be treated with aprotic solvent condition without light. The discussion on this point was added in the manuscript, lines 118-124 and 367-371.
Try to discuss the anti-inflammatory findings with a possible bioactive compounds mode of action against the tested cell lines.
Include future trends to keep working with the obtained data.
: In this study, we tried to isolate unstable metabolites from the culture broth of Fusarium sp. QF001. Some portion of isolates were lost during the isolation. Due to the limited amount of the isolated compound, we investigated toxicities and basic anti-inflammatory activities. As the extension of this research, the compounds revealed as anti-inflammatory could be purified more and they can provide the chance to inspect the detail mode of action of the compounds. Also, the interaction between endophytic fungus and host-plant in the aspects of secondary metabolites and pharmacological use of host plant can be studied in the future research.
Include a statistical description in the figures that required it.
: Thank you for valuable comments. We have revised the cell viability graph (Figure 5) and the associated figure legend.
Try to conclude with a general statement of the most relevant part of this study.
: General conclusion including future study was added to the manuscript, lines 367-374.
Round 2
Reviewer 1 Report
The authors earnestly revised the manuscript. I believe this manuscript will be suitable for publication in Molecules.
Reviewer 3 Report
I appreciate the changes on the new version submittend by the authors.
The first version seems to me have a good English level (but i am not an native English speaker). However, it seems that in this new text some errors remains
eg : were previously reported with the cell cycle inhibitor or cytotoxins
200 against cancer cells
eg : and the all the isolates did not show cytotoxicity
and some others.